# Predicting Empathy and Other Mental States During VR Sessions Using Sensor Data and Machine Learning

**DOI:** 10.3390/s25185766

**Published:** 2025-09-16

**Authors:** Emilija Kizhevska, Hristijan Gjoreski, Mitja Luštrek

**Affiliations:** 1Institut “Jožef Stefan”, 1000 Ljubljana, Slovenia; mitja.lustrek@ijs.si; 2Jožef Stefan International Postgraduate School (IPS), 1000 Ljubljana, Slovenia; 3Faculty of Electrical Engineering and Information Technologies, Ss. Cyril and Methodius University, 1000 Skopje, North Macedoniahristijang@feit.ukim.edu.mk; 4Emteq Labs, Brighton BN1 9RS, UK

**Keywords:** empathy, machine learning, mental states, sensor data, virtual reality

## Abstract

Virtual reality (VR) is often regarded as the “ultimate empathy machine” because of its ability to immerse users in alternative perspectives and environments beyond physical reality. In this study, 105 participants (average age 22.43 ± 5.31 years, range 19–45, 75% female) with diverse educational and professional backgrounds experienced three-dimensional 360° VR videos featuring actors expressing different emotions. Despite the availability of established methodologies in both research and clinical domains, there remains a lack of a universally accepted “gold standard” for empathy assessment. The primary objective was to explore the relationship between the empathy levels of the participants and the changes in their physiological responses. Empathy levels were self-reported using questionnaires, while physiological attributes were recorded through various sensors. The main outcomes of the study are machine learning (ML) models capable of predicting state empathy levels and trait empathy scores during VR video exposure. The Random Forest (RF) regressor achieved the best performance for trait empathy prediction, with a mean absolute percentage error (MAPE) of 9.1%, and a standard error of the mean (SEM) of 0.32% across folds. For classifying state empathy, the RF classifier achieved the highest balanced accuracy of 67%, and a standard error of the proportion (SE) of 1.90% across folds. This study contributes to empathy research by introducing an objective and efficient method for predicting empathy levels using physiological signals, demonstrating the potential of ML models to complement self-reports. Moreover, by providing a novel dataset of VR empathy-eliciting videos, the work offers valuable resources for future research and clinical applications. Additionally, predictive models were developed to detect non-empathic arousal (78% balanced accuracy ± 0.63% SE) and to distinguish empathic vs. non-empathic arousal (79% balanced accuracy ± 0.41% SE). Furthermore, statistical tests explored the influence of narrative context, as well as empathy differences toward different genders and emotions. We also make available a set of carefully designed and recorded VR videos specifically created to evoke empathy while minimizing biases and subjective perspectives.

## 1. Introduction

Empathy is a complex and multifaceted concept that has captured the interest of researchers in various disciplines, including psychology, neuroscience, philosophy, sociology, and anthropology. Despite the absence of a universally accepted definition, empathy is widely understood as a multidimensional construct that encompasses both cognitive and affective components [1,2]. The cognitive component refers to the ability to construct a working model of another person’s emotional state, a process conceptually related to Theory of Mind (ToM) yet distinct in that it centers on understanding emotions rather than beliefs or intentions. In contrast, affective empathy involves sensitivity to and the vicarious experience of another person’s emotions, without necessarily sharing them directly [3]. In this context, the present study is among the first to uniquely combine immersive VR experiences with multimodal physiological sensing and machine learning in order to predict empathy. Examining the reasons why empathy is a valuable area of study highlights its importance across various fields, such as (1) empathy in social relationships and cultures: empathy enables individuals to understand others’ emotions, fostering interpersonal connections, and its cultural variations provide insights into cross-cultural communication and understanding [4,5,6]; (2) prosocial behavior and altruism: empathy motivates prosocial actions such as helping, volunteering, and altruistic behavior, contributing to societal well-being [7,8]; (3) mental health: empathy deficits are linked to psychiatric disorders such as autism and antisocial personality disorder, highlighting its importance in improving diagnostic and therapeutic approaches [4]; (4) technological applications: advances in VR and AI have enabled empathy-driven technologies that enhance user interaction and communication by creating immersive, perspective-shifting experiences [9]; etc.

Although no universally accepted “gold standard” exists for measuring empathy [10], several established methods are widely used in research and clinical contexts. These include self-report questionnaires, behavioral observation, psychophysiological measures, and performance tasks.

Self-report questionnaires: These involve participants answering questions about their thoughts, feelings, and behaviors related to empathy. While they provide valuable insights into subjective experiences, they rely on self-perception and may be influenced by biases such as social desirability or inaccurate self-assessment [6,11].Performance tasks: These tasks evaluate empathic abilities by having participants engage in simulations or scenarios that elicit perspective-taking or emotion inference. They offer ecologically valid measures but primarily capture cognitive empathy and may not reflect emotional empathy. Task design and individual cognitive differences can also influence results [12].Behavioral observation: This method involves recording empathic behaviors, such as facial expressions, vocal tone, or prosocial actions, in natural or controlled settings and using standardized coding schemes for quantifying these behaviors. While it provides objective data, it can be time-consuming and subject to reactivity biases due to the presence of observers. Additionally, it focuses on behavioral expressions rather than emotional empathy [13].Psychophysiological measures: These assess physiological responses, such as heart rate, skin conductance, or brain activity, during empathic experiences. Although they do not directly predict empathy, techniques like Functional Magnetic Resonance Imaging (fMRI) and electroencephalography (EEG) have been used to study the neural mechanisms of empathy, highlighting the roles of the mirror neurons and brain regions responsible for emotional processing. However, these methods may be influenced by non-empathetic factors, inter-individual variability, and technical constraints [14,15]. Importantly, measures such as heart rate (PPG) and muscle activity (EMG) have been associated with affective arousal, while movement dynamics captured by an IMU can serve as indirect markers of cognitive engagement, thereby aligning these modalities with both affective and cognitive components of empathy.

The choice of method depends on the research goals, the population studied, and the empathy dimensions targeted. To address the limitations of individual approaches, researchers often combine multiple methods to capture empathy comprehensively, despite the added complexity and time required. An ideal empathy measure would be objective, capturing subtle responses that might not be visible through researcher observation, and applicable during arbitrary tasks. Psychophysiological measures best meet these criteria, despite the fact that the precise connection between these measures and empathy remains an open question [16].

VR has proven effective in enhancing emotional engagement and empathy across various fields and is increasingly used as a tool for empathy training [17,18]. VR is a technology that generates computer-simulated realistic environments, allowing users to immerse themselves and interact within a virtual setting. By utilizing a VR headset, and, in some cases, additional devices, users can experience a sense of being present in a simulated world, engaging with virtual elements in an immersive and interactive manner [19,20]. As a result, many studies have emphasized VR’s capacity to elicit empathic responses, often describing it as “the ultimate empathy machine” [21].

Some of the main reasons why VR is referred to as “the ultimate empathy machine” are as follows:Immersive and embodied experiences: VR creates a sense of presence, allowing users to embody different perspectives and experiences. This immersion can enhance empathy by enabling individuals to see and feel situations from another person’s viewpoint [22] or a different viewpoint [23]. For instance, one study introduced a method where participants, seated in a stationary chair and wearing a VR headset, used a joystick to move an avatar forward in a virtual environment, allowing them to experience a different point of view. This approach employed electrical muscle stimulation to synchronize leg muscle sensations with the gait cycle, offering a first-person perspective that created the illusion of walking without physical movement. The study demonstrated that this technique, through VR immersion, effectively induced an embodied experience with the sensation of leg movement for seated users [23].Perspective-taking and emotional engagement: VR can simulate realistic scenarios that elicit emotional responses and encourage perspective-taking. VR experiences can simulate lifelike scenarios that elicit emotional responses and immerse users in the perspectives of others, helping them better understand different feelings and experiences [24]. For example, one study demonstrated how VR can enhance emotional well-being and foster empathy in the elderly by exposing them to emotionally meaningful content. This included storytelling, immersive environments, and virtual tours of nostalgic locations. Simulated social interactions with family or peers were also incorporated to encourage emotional engagement [25]. Another notable example is the use of death simulations in VR, where users explore the concept of death in a controlled environment. As a profound and universal theme, death offers unique opportunities for fostering empathy, emotional development, and self-reflection. In these simulations, participants transition from a virtual cardiac arrest to brain death, experiencing inner-body sensations while journeying through expansive cosmic landscapes. This immersive experience encourages deep contemplation on mortality and the sublime nature of existence, highlighting VR’s capability to engage with complex emotional and existential topics [26].Empathy training and perspective shifts: VR-based interventions are used in healthcare, education, and diversity training to challenge biases and enhance empathic abilities. By immersing individuals in realistic scenarios, VR challenges biases and fosters perspective-taking, allowing participants to gain deeper insights into others’ emotions and experiences, making it a powerful tool for empathy training [17]. For example, VR has been shown to effectively cultivate empathy in sensitive areas such as racism, inequity, and climate change, particularly within healthcare, where participants engage in transformative experiences that improve empathy and understanding [27]. In another study, VR-based dementia training for home care workers improved their knowledge, attitudes, competence, and empathy, demonstrating the value of immersive training methods in enhancing care skills [28]. Additionally, a study proposed a video game framework using generative AI to create personalized interactive narratives, aiming to foster empathy and social inclusion. By combining VR’s immersive nature with personalized experiences, this research explored how AI-driven games can enhance wisdom, with a particular focus on empathy as a core element of wisdom [29].Ethical considerations: Although VR has the potential to enhance empathy, its use must be guided by ethical principles. As a tool capable of evoking intense emotional responses, VR experiences should be designed responsibly to respect the dignity and privacy of all individuals involved. Additionally, it is crucial to carefully manage the emotional impact on users to avoid potential negative effects on their well-being. Achieving a balance between creating immersive empathic experiences and ensuring the emotional safety of participants is essential for using VR effectively as a tool to foster empathy [27]. For instance, scenarios like virtual rape highlight the need to address how VR experiences might awaken the idea of real-world violations [30]. Another study emphasized that VR can unintentionally cause harm, including empathy fatigue or distress for users and providers of empathic acts, especially in contexts involving chronic or stigmatized illnesses. Thoughtful design approaches that balance emotional engagement with actionable support are vital to ensure VR enhances well-being without adverse effects. By integrating strategies to mitigate empathy fatigue and promote self-care among users and caregivers, VR technologies can better align with the goal of positively impacting health and well-being [31].

Despite growing interest in both VR applications and empathy research, there remains a notable gap in integrating physiological sensing with immersive VR to objectively model empathic processes. To address this gap, the objective of this study was to explore the relationship between participants’ different types of empathy and changes in their physiological responses, measured by sensors such as an inertial measurement unit (IMU), a photoplethysmograph (PPG), and an electromyograph (EMG). Participants were immersed in 360° 3D VR videos that we created and recorded, wherein actors expressed various emotions (sadness, happiness, anger, and anxiety). Afterward, they filled out brief questionnaires to assess their levels of state and trait empathy, as well as arousal, valence, and discomfort. Using their sensor and questionnaire data, ML models were developed that predict a participant’s empathy score based on their physiological arousal while watching the VR video.

Developing a stable and accurate model to predict individuals’ empathy levels could provide several advantages:Personalized interventions and early prevention: The model can assess individuals’ empathy levels across different contexts, such as healthcare, education, or counseling. This information could guide strategies to foster empathic skills in those who may benefit from additional support and help prevent potential challenges in settings where empathy is critical, such as patient care or interpersonal relationships.Selection and training: It can help select individuals for empathy-intensive roles and guide training programs by identifying areas for improvement.Research and understanding: The model can contribute to research on empathy, offering insights into factors influencing it and identifying patterns across populations.Entertainment and interactive media: By tailoring content to users’ empathy levels, creators can enhance emotional engagement in video games, narratives, and media recommendations.Personal growth and self-awareness: Individuals can gain self-awareness about their empathy strengths and weaknesses, fostering personal growth and encouraging the development of empathy.

Overall, a stable model for predicting empathy could impact personalized interventions, selection and training, research, entertainment, and personal growth [4,32,33].

This study also aimed to develop ML predictive models that predict empathic arousal and non-empathic arousal, and distinguish between empathic and non-empathic arousal, as well as between relaxed and aroused states. Additionally, statistical tests were conducted to determine whether participants showed more empathy towards females or males, whether participants displayed greater empathy toward certain emotions compared to others, and whether participants exhibited greater empathy when they understood the reasons behind others’ emotions.

## 2. Background and State of the Art

Recent research highlights the role of emotion regulation strategies, such as mindfulness, self-compassion, and resilience, as significant predictors of well-being [34]. A decline in empathy has been associated with increased psychological distress [35], whereas higher levels of empathy are linked to improved well-being, reduced burnout, and more meaningful work experiences [2,36]. Given the substantial body of research suggesting that VR can be used to elicit empathic behavior [21], the recognition of psychophysiological measures as objective indicators of emotional states [14,15], and the established capability of ML to analyze large datasets, identify patterns, and make predictions, this section examines the recent research in four areas:

Section 2.1 explores how VR has been utilized to enhance empathy, the contexts in which it has been applied, and the populations involved.

Section 2.2 reviews findings from studies investigating the most effective environments—VR-based or not—for eliciting empathy, along with technical factors influencing effectiveness.

Section 2.3 examines the measures and methodologies employed in the existing research to assess empathy.

Section 2.4 addresses a gap in the literature; to our knowledge, no studies have used VR to elicit empathy, measured it through sensor-based approaches, and developed predictive ML models. Therefore, we review studies on empathy and other mental state predictions, focusing on how they employed sensor data and developed ML models.

### 2.1. General Context of the Studies Using VR in Empathy Enhancement

Investigations have explored VR’s capacity to foster empathy across various contexts. In healthcare, VR has been employed to enhance empathy and deepen the understanding of the challenges faced by patients with dementia among healthcare workers [37] and caregivers [38], promote empathy towards individuals with psychotic disorders among mental health professionals [39], and enhance medical students’ empathy towards patients with depression [40], opioid use disorder [41], and Myalgic Encephalomyelitis/Chronic Fatigue Syndrome (ME/CFS) [42]. VR has also been applied to undergraduate healthcare students to simulate experiences of older adults with cognitive impairments, disorientation, or distorted sensory perceptions [43].

Beyond healthcare, VR has been used to promote prosocial behavior and empathy towards peers [44], homeless individuals [9], refugees [27], victims of sexual harassment [45], avatars experiencing pain or pleasure [46,47,48], and groups facing challenges related to pregnancy, aging, wheelchair use, or visual impairments [49,50].

These diverse applications demonstrate that VR has primarily been developed as a tool to improve empathy in specific contexts, such as environmental, healthcare, and prosocial scenarios, rather than as a general tool for measuring empathy. Overall, these studies highlight VR’s potential to foster context-specific empathic understanding across different domains.

### 2.2. Using Virtual Reality to Elicit Empathy

Studies investigating the technical and environmental aspects of VR have consistently shown that 360º immersive virtual environments (IVEs) are more effective in eliciting empathy than two-dimensional videos or films [27,49,51]. Other forms of content, such as e-course materials [38] and workshops [37], showed smaller effects. Evidence also suggests that certain populations, including older participants and non-native English speakers, benefit more from immersive experiences. Narrative-based perspective-taking exercises further enhance empathy, particularly when presented before the VR experience [9,45].

Pretest–posttest designs have confirmed these effects; immersive VR increased empathy toward individuals with opioid use disorder [41], older adults with cognitive impairments [43], and people living with ME/CFS [42]. Other interventions simulated psychotic disorder symptoms, showing higher user satisfaction despite non-significant empathy changes [39], while 3D VR simulations of daily life for patients with depression effectively fostered empathy compared to standard medical education [40].

Technical factors also influence outcomes: First-person perspectives increased ownership feelings compared to third-person views [47,48]. Interacting with familiar avatars expressing pain, or taking the perspective of someone expected to be encountered, enhanced empathic responses [44,46].

Overall, these studies indicate that VR can elicit empathy effectively, but the research largely relies on self-reports and seldom combines VR with multimodal physiological measurements, leaving the gap that our study addresses.

### 2.3. Measuring Empathy

Most studies evaluate empathy using self-reported questionnaires, including single- or multi-item instruments such as the Interpersonal Reactivity Index [9,39,40,47] and its subscales [37,48], the Empathy Scale [45], the Jefferson Scale of Empathy (JSE) [40,41], the Kiersma–Chen Empathy Scale (KCES) [43], the Situational Empathy and Perspective-Taking Scale [37], and the Empathy for Pain Scale [47,48], as well as adapted items from other tools [27,38,42,49,50].

Objective measures have also been used, including galvanic skin response (GSR) [46], skin conductance reactivity (SCR) [47,48], heart rate (HR) [47], EEG [49], and motion tracking via IMU or infrared-based systems [9,46]. While these methods provide complementary insights, statistically significant findings have been limited, and most measures have not directly linked physiological signals to both affective and cognitive components of empathy.

Some studies, including the fuzzy logic–based estimation study [52], monitored physiological signals across multiple sessions to track how participants’ emotional and cognitive states evolved over time, but ML was not applied. These investigations highlight the dynamic nature of empathy and the value of repeated measurements for understanding individual differences.

In our protocol study, we carefully selected PPG, EMG, and IMU sensors based on their ability to capture the arousal, emotional expression, and movement patterns associated with empathy, respectively [53]. Here, we provide a concise explanation to support their relevance for predicting empathic responses in combination with ML.

### 2.4. Predicting Empathy and Other Psychological States with Machine Learning

Outside VR contexts, physiological signals have been combined with ML to predict empathy and related states. Examples include monitoring healthcare professionals’ mood, well-being, and empathy via smartwatches and self-reports over the course of months [54], applying neural networks to brain imaging [55], and linking genetic or behavioral data to emotional responses [56]. Physiological signals such as EEG, PPG, EMG, and EDA have been used with classifiers like the SVM, kNN, CNN, LSTM, and Extreme Learning Machines to predict emotional states or flow experiences [57,58,59,60,61,62,63,64].

VR-based studies have similarly employed wearable sensors during immersive experiences to predict psychological states such as arousal, stress, or valence [65,66,67,68]. While these approaches have proven effective for intra-subject classification, inter-subject generalization remains limited. Taken together, they establish a strong methodological foundation. However, despite their success in predicting other psychological states, comparable methods have not been applied to empathy. This gap underscores the novelty of our work, which uniquely combines immersive VR, multimodal physiological sensing, and ML to predict empathy.

## 3. Materials and Data Collection Process

### 3.1. Materials and Setup for Empathy Elicitation in VR

Based on the findings in Section 2.2, we immersed participants in a 360º 3D virtual environment, as VR has proven highly effective in eliciting empathic responses. Comparative analyses indicate that 360º settings are as effective as or more effective than other approaches, with both immediate and lasting impacts. The use of 3D immersion, as highlighted by the review [2], is superior to two-dimensional experiences. Participants adopt a first-person perspective, observing actors expressing emotions directly in front of them, which research shows is more effective for empathy than a third-person perspective [47,48].

We decided to create and record videos where actors genuinely express four distinct emotions (happiness, sadness, anger, and anxiousness) without content, words, or explanations to avoid confounding empathy with emotional responses to the content (videos are sent as Appendix A to the paper) [27]. Moreover, individuals may display varying empathic reactions depending on the emotional valence, which led to the inclusion of four distinct emotions in our study [69]. In the non-narrative version, all four actors portrayed the four emotions by gradually intensifying each emotion to its peak and then returning to a neutral state, without accompanying narratives. For participants who may find understanding the reasons behind emotions more impactful, we developed a narrative version of the VR session featuring emotional audio narratives in Slovenian, followed by corresponding videos (50–120 s). We created two narrative versions. For each narrative version, to ensure gender balance, we recorded one video with two male actors and one video with two female actors [53]. Informed consent was obtained from the actors prior to the recording. Each version consisted of four parts, each addressing one emotion, with consistent themes of childhood abuse, abandonment, and mother–child relationships (Figure 1).

Prior to the VR session, participants completed questionnaires on demographic and personal details, including age, gender, health status, language proficiency, prior VR experience, and educational background [53]. Additionally, the Questionnaire of Cognitive and Affective Empathy (QCAE) [3] was administered to assess participants’ trait empathy, which reflects an individual’s innate capacity for empathetic behavior and is closely tied to personal characteristics [70]. In this study, we adopt the following definition of empathy: the ability or tendency to form an internal model of others’ emotional states and to be sensitive to and share their feelings while remaining aware of the distinction between the self and others [3]. The affective empathy subscale of the QCAE was later used as a ground truth in our analysis.

The VR session began with a calibration phase, where participants relaxed for two minutes while avoiding facial expressions and head movements. This was followed by voluntary facial expressions, which provided subject-specific normalization for EMG signals based on established methods [71,72,73]. Then, there was a two-minute forest video, ‘The Amsterdam Forest in Springtime’, and its purpose was to establish a baseline representing participants’ relaxed state (Figure 2).

In the main segment, participants experienced four emotional states (anger, sadness, happiness, and anxiousness) through narrative–video pairs or videos alone. In the narrative condition, each video was preceded by a corresponding emotional narrative read by the actor, creating a 2–3 min immersive experience for each pair. While the segments in the narrative versions followed stories, the segments in the non-narrative versions were counterbalanced to prevent bias from order effects [53].

After each pair, participants reported their current empathic state using the 2nd, 3rd, and 4th items from the 12-item State Empathy Scale [74], selected to focus on affective empathy while excluding questions unrelated to empathy. Additionally, the question period served to reduce habituation and prevent emotional carry-over between segments. This process was repeated for all four emotions, with each feedback session lasting approximately 30 s. Participants also rated their arousal and valence using the Self-Assessment Manikin (SAM) affective rating system [75], alongside a measure of personal distress derived from the Interpersonal Reactivity Index (IRI) [11]. These additional measures provide a broader understanding of emotional responses and their connection to empathy [76].

To differentiate empathic arousal from general arousal, a roller coaster video (Yukon Striker, Canada’s Wonderland) was shown as a control. The intensity of arousal and valence after the control video were compared to responses from the narrative–video pairs, strengthening the study’s validity.

The session concluded with a presence questionnaire adapted from the Bongiovi Report [53] modified to include questions about participants’ VR experiences and potential challenges [27]. Following recommendations for minimizing VR sickness, fatigue, and potential health concerns, the entire session, including calibration and control measures, lasted approximately 20 min [53]. Participants were randomly assigned in equal numbers to one of five versions—two narrative scenarios, each performed separately by male and female actors, and one non-narrative version—ensuring a balanced design suitable for diverse linguistic backgrounds.

### 3.2. Participants and Recording Setup

In this study, we employed convenience sampling to recruit participants from the general population, including university students, employees at our institute, and the wider public, with invitations distributed both verbally and in writing. Exclusion criteria were defined to ensure both participant safety and the reliability of physiological measurements. Individuals under 18 years of age were excluded due to ethical considerations and the developmental differences in empathic processing. Participants with epilepsy, heart conditions, or disorders affecting the autonomic nervous system were excluded to avoid the health risks associated with immersive VR exposure and physiological arousal. Individuals with diagnosed anxiety disorders (e.g., generalized anxiety disorder, panic disorder, social anxiety disorder, or post-traumatic stress disorder) were excluded because these conditions could introduce confounding effects on stress- and emotion-related physiological measures. In addition, individuals with uncorrected vision problems that prevented the proper use of VR headsets were excluded. Inclusion criteria required participants to be adults willing to engage in scientific research with normal or corrected-to-normal vision (minor refractive errors or corrective lenses acceptable). The final sample comprised 105 participants, with a mean age of 22.43 ± 5.31 years (range: 19–45), of whom 75.24% identified as female. While established methods exist for power analysis in classical statistical modeling, determining the precise sample size requirements for machine learning remains challenging. Based on comparable studies [38,45,46,47,48,77] which successfully developed predictive models with smaller samples, we considered our sample size sufficient for the present study. We acknowledge the gender imbalance as a limitation with respect to the generalizability of the findings.

Ethical approval for the study was obtained from the Research Ethics Committee at the Faculty of Arts, University of Maribor, Slovenia (no. 038-11-146/2023/13FFUM), as the study posed no physical, psychological, legal, or social risk to the participants. In line with these ethical standards, the complete dataset was securely maintained on the servers of the research institute in Slovenia (Jožef Stefan Institute). To protect participants’ privacy, access control protocols were put in place by our team. Only authorized personnel could access the study data, either through the local network or via a Virtual Private Network (VPN). All participant information was pseudonymized by assigning a unique, randomly generated numeric ID in place of full names. The correspondence between these IDs and participants’ actual identities was stored separately from the main dataset, adding an extra layer of security. At participants’ requests, their data were removed, and, upon publication, the dataset will be fully anonymized by eliminating the link between IDs and identities. The data from the participants were collected using the emteqPRO system, which includes a sensing mask attached to the Pico Neo 3 Pro Eye VR headset (Figure 3). The system provides both raw sensor data and derived variables through its Emotion AI Engine, which leverages data fusion and ML techniques to assess the user’s emotional state, as described in the protocol study paper [53]. A total of 27 derived variables were generated for each recording, grouped into 9 categories: 7 features related to heart rate variability (HRV) and 3 related to breathing rate (all derived from PPG), 2 features representing facial expressions (from EMG), 4 for arousal (from EMG and PPG), 4 for valence (from EMG and PPG), 1 for facial activation and 1 for facial valence (both from EMG), 1 for head activity—defined as the percentage of time with detected head movement (from an IMU)—and 4 features representing muscle activation from dry EMG electrodes on the zygomatic, corrugator, frontalis, and orbicularis muscles, expressed as a percentage of the maximum activation recorded during calibration [73].

These derived features were provided at varying intervals, from 10 s to 500 ms, resulting in different data frequencies. Prior to preprocessing, the total number of instances from all participants exceeded 136.5 million.

## 4. Methodology

### 4.1. Preprocessing

The emteqPRO system generated a separate file for each participant. As the first preprocessing step, duplicates in both columns and rows were removed for each participant, reducing the total number of features to 23, since 4 features had identical values.

Since all features were numeric except for one categorical feature, “Expression/Type”, which had three categories—smile, frown, and neutral—we applied one-hot encoding to this feature. One-hot encoding is a preprocessing technique that converts categorical variables into a numerical format by creating separate binary (0 or 1) features for each unique category.

Next, missing values, which accounted for less than 1% of each participant’s total data, were imputed using the mean value of the respective feature for that participant. We also removed rows and columns containing repeated or identical values to avoid redundancy and ensure data quality. Finally, we used a StandardScaler to normalize the data by transforming each feature to have zero mean and unit variance, ensuring that all features contributed equally to the analysis. However, normalization was performed individually for each participant rather than for the entire dataset. This approach ensured that data from participants with generally lower physiological attribute values in a relaxed state were not overshadowed or deemed less significant.

### 4.2. Data Segmentation and Feature Engineering

Since some participants requested that specific questions be repeated after certain narrative videos, and because the initial calibration process took longer for participants who experienced discomfort with the VR headset, the session timetables varied among participants (Figure 4). Based on each participant’s timetable and the video they were watching, the data for each participant were split into six segments, labeled according to specific parts of the videos: one for the forest video, four for each emotion-related video segment, and one for the roller coaster segment.

Given that features were recorded at intervals ranging from 1 s to 500 ms, the data were further divided into three time windows: 500 ms, 5 s, and one entire video segment (each video consisting of six segments/parts). For each original feature or insight, except for the “Expression/Type” feature (which was restored to its original categorical form), the minimum, maximum, average, and standard deviation were calculated. Additionally, for the “Expression/Type” feature, the majority value was computed within each window segment, resulting in a total of 89 features.

As the final step of feature engineering, additional features were introduced. These included responses from the personal information questionnaire and the presence questionnaire for each participant. Furthermore, features derived from the QCAE [3] were incorporated, such as perspective-taking, online simulation, affective empathy, emotion contagion, and proximal responsivity. Additionally, responses related to valence, arousal, personal distress, and the three questions assessing state empathy were included [53].

## 5. Experimental Evaluation

### 5.1. Experimental Setup

#### 5.1.1. Conducted Experiments

Different segments, using 500 ms and 5 s window sizes: We set up five different experiments, labeling the dataset in five different ways: (a) binary classification of empathic arousal prediction: this aimed to predict empathic arousal by comparing empathic parts of the video with the forest part, excluding the non-empathic roller coaster segment; (b) binary classification of non-empathic arousal prediction: this compared the forest and roller coaster parts of the video to predict non-empathic arousal; (c) binary classification of empathic vs. non-empathic arousal: this included only empathic parts and the roller coaster, aiming to distinguish between empathic and non-empathic arousal while analyzing physiological responses to empathic content and non-empathic arousal-inducing stimuli (e.g., the roller coaster); (d) binary classification of general arousal prediction: this split the dataset into two classes: the forest (representing no arousal) and everything else (including empathic parts and the roller coaster, representing arousal); (e) three-class classification of arousal type prediction: this treated the forest and roller coaster as separate classes while grouping all empathic parts into one class, aiming to distinguish no arousal, empathic arousal, and non-empathic arousal without differentiating specific empathic emotions [53].State empathy as nominal classification, using 500 ms and 5 s window sizes: We aimed to predict state empathy, which reflects the temporary affective response elicited in specific situations [78]. This approach used participants’ averaged responses to the three state empathy questions for each video segment. Each empathic segment was treated separately and labeled with the corresponding state empathy responses, while the forest and roller coaster segments were labeled as zero. The goal was to predict participants’ state empathy levels during the session.State empathy as nominal classification, using the ‘one entire video segment’ window size: The target variable was obtained as an average of participants’ responses to the state empathy questions provided after each segment of the video session. However, in this case, when the ground truth was the state empathy class, we used a window size corresponding to one entire video segment (with each video consisting of six segments/parts). This window size was used consistently in all subsequent experiments.State empathy as ordinal classification, using the ‘one entire video segment’ window size: Since state empathy has ordered categories, ranging from not empathic to highly empathic, and undefined distances between levels, we applied ordinal classification.Trait empathy as regression, using the ‘one entire video segment’ window size: Since trait empathy was a continuous variable derived from the continuous attributes of cognitive empathy (perspective-taking and online simulation) and affective empathy (emotion contagion and proximal responsivity) from the QCAE, we focused on predicting the affective empathy component. Affective empathy reflects the ability to feel and share another’s emotions, which aligns with the context of this study, as the participants were not required to perform actions that would test cognitive empathy, such as understanding the actors’ feelings.Trait empathy as classification, using the ‘one entire video segment’ window size: Equal interval binning was applied to the target variable, trait empathy (affective points), dividing its range (5.00 to 8.75) into equal intervals. The resulting bins were defined as follows: class 0: [5.00–5.75); class 1: [5.75–6.50); class 2: [6.50–7.25); class 3: [7.25–8.00); and class 4: [8.00–8.75].

#### 5.1.2. Resampling Approaches

For schemes (b), (c), and (e), related to the different segments outlined in Section 5.1.1, where the datasets were imbalanced (with the majority class accounting for 80% of the samples), we applied four resampling approaches for each developed model: (1) the Synthetic Minority Oversampling Technique (SMOTE): this method generates synthetic samples for the minority class to balance the dataset; (2) RandomUnderSampler (RUnderS): this method reduces the majority class by randomly selecting samples to balance the dataset; (3) SMOTETomek: this technique combines SMOTE for oversampling the minority class with the Tomek links method for undersampling the majority class, effectively addressing both class imbalances; and (4) the unaltered dataset: models were trained on the original dataset without applying any sampling techniques.

#### 5.1.3. Machine Learning Algorithms

We developed various models to predict participants’ state empathy classification and affective empathy classification, and to predict different segments during the VR session. For the classification tasks, we employed seven algorithms representing different methodological approaches: a probabilistic model (Gaussian Naive Bayes), a linear model (Stochastic Gradient Descent), an instance-based model (K-Nearest Neighbors, KNN), ensemble tree-based models (the Random Forest classifier and Extreme Gradient Boosting, XGBoost), a single tree-based model (the Decision Tree Classifier), and a generalized linear model (Logistic Regression). This selection ensures coverage of diverse learning paradigms, enabling systematic comparison and identification of the most effective modeling strategies for predicting empathy-related states from our dataset.

For predicting participants’ affective empathy through regression, we developed models using seven different regression algorithms: the RF Regressor, Decision Tree Regressor, Gradient-Boosting Regressor, XGBoost Regressor, Support Vector Regressor, K-Nearest Neighbors Regressor, and Elastic Net Regressor.

Additionally, for predicting participants’ state empathy as an ordinal classification problem, four algorithms were employed: Ordinal Random Forest (ORF), Ordinal SVM (OSVM), Ordinal Gradient Boosting (OGB), and Ordinal Logistic Regression (OLR).

#### 5.1.4. Cross-Validation Strategy

Model evaluation was conducted using a Leave-One-Subject-Out cross-validation (LOSO-CV) approach. In this method, each participant, identified by their unique ID, was excluded as a test set to assess the model’s generalization performance.

#### 5.1.5. Evaluation Metrics

For classification, the balanced accuracy score was used to evaluate model performance, calculated as the average recall across all classes, ensuring a fair assessment in the case of imbalanced datasets. Additionally, confusion matrices were employed to compare actual and predicted labels, providing further insights into model performance. To quantify variability in classification results across folds, the standard error of the proportion (SE) was calculated using the following formula:$$SE=\sqrt{\frac{p(1-p)}{n}}$$
where *p* is the proportion of correctly classified instances (e.g., balanced accuracy) and *n* is the number of observations or folds. This measure allows the reporting of classification results as p±SE, providing an estimate of uncertainty around the mean performance [79].

For regression, models were evaluated using two key metrics: the mean absolute percentage error (MAPE) and the symmetric mean absolute percentage error (SMAPE). The MAPE quantifies the average percentage difference between predicted and actual values, offering an intuitive measure of prediction accuracy relative to the data’s scale. In contrast, the SMAPE measures the percentage error by considering both the predicted and actual values together. This symmetric approach reduces any bias from over- or under-predictions, providing a more balanced assessment. The variability of regression results across folds is expressed using the standard error of the mean (SEM), calculated as follows:$$SEM=\frac{\sigma }{\sqrt{n}}$$
where σ is the standard deviation of the metric across folds and *n* is the number of folds. Reporting regression results as mean±SEM allows a clear representation of uncertainty in the predictions [79].

### 5.2. Predicting Different Segments

A comprehensive analysis was conducted by developing models using seven classification algorithms on two window sizes (500 ms and 5 s). These models were evaluated using four data-balancing techniques—undersampling, oversampling, a combination of techniques, and the original dataset—across five different segments (labeling schemes). This resulted in a total of 280 unique confusion matrices and their corresponding accuracies for every combination (Figure 5).

Regarding the experiments on different segments (labeling schemes), we can conclude the following: (1) empathic arousal could be predicted with relatively well-balanced confusion matrices and high accuracy across most models; (2) non-empathic arousal was also reliably predicted, with almost all models achieving a balanced accuracy exceeding 60% and some reaching up to 78 ± 0.63% SE, along with a reasonable balance across classes; (3) distinguishing between empathic and non-empathic arousal was possible, with balanced accuracy reaching 79 ± 0.41% SE; (4) general arousal prediction demonstrated similarly high accuracy and class balance; (5) differentiating between no arousal, empathic arousal, and non-empathic arousal showed moderate success. These results align with prior VR studies that demonstrated the ability to distinguish emotional arousal states from physiological signals, although our approach extends previous work by including multiple VR content types and ML [41,42,46].

Regarding the two window sizes, both window sizes demonstrated similar class balance and balanced accuracy, but the 5 s interval dataset performed slightly better (Figure 5 and Figure 6).

Regarding the data-balancing techniques, the undersampling method yielded the best results for the dataset extracted at 5 s intervals, while the SMOTETomek produced a similar or slightly lower performance. For the dataset extracted at 500 ms intervals, the SMOTE achieved the best results, followed closely by the SMOTETomek and the original dataset. Overall, the SMOTETomek consistently provided the best or second-best performance across all combinations of labeling schemes [53].

Regarding the classification algorithms in this case, the Gaussian Naive Bayes exhibited the poorest performance, particularly in achieving balanced confusion matrices, while the RF classifier and XGBoost emerged as the top-performing algorithms across all combinations, with the RF classifier slightly outperforming XGBoost in most segments (Figure 5 and Figure 6) [53].

### 5.3. Predicting State Empathy

#### 5.3.1. State Empathy Level as Nominal Classification, Using 500 ms and 5 s Window Sizes

The regression outputs were converted into classes by dividing the continuous scale into intervals centered around integer values (for example, values from 0.5 to 1.4 were assigned to class zero and those from 1.5 to 2.4 to class one, continuing similarly up to class four). Using the window sizes of 500 ms and 5 s, it was very challenging to predict the precise level of empathy participants were feeling during the session, and to determine whether they were empathizing by mirroring emotions or experiencing something different while observing specific emotions. Balanced accuracy for this task reached only 28 ± 0.36% SE, with confusion matrices revealing imbalances across multiple classes (Figure 5 and Figure 6) [53].

#### 5.3.2. State Empathy as Nominal Classification, Using the ‘One Entire Video Segment’ Window Size

Although the State Empathy Scale we used in this study contains six grades, no average was calculated for the three questions that received a score of 0, resulting in five distinct classes (0–4) [53].

To enhance the classification process, box plots were generated to reduce the number of classes and identify optimal merges. The plots were used to examine the relationship between the five most important features and the target variable, the state empathy class. These features were selected based on RF feature importance, which identified the following five most influential features: the standard deviation of the mean heart rate variability; the valence class feature, categorized as −1 (negative), 0 (neutral), or 1 (positive); the motion intensity recorded by the IMU sensor; the activation percentage of specific muscles, particularly the frontal muscles, relative to their maximum activation during the calibration session; participants’ self-report for discomfort levels experienced during the VR session after each part of the video.

The analysis of all box plots revealed a clear strategy for merging based on the distributions of features. The box plots suggest that classes 1, 2, 3, and 4 exhibit substantial overlaps in ranges, medians, and variability, which could support merging them into a single group. However, to retain meaningful distinctions, we created three classes by merging classes 1 and 2 and classes 3 and 4 and keeping class 0 separate. This balanced simplification and interpretability. The reasoning is as follows: (1) class 0 stands out with distinct medians, non-overlapping ranges, and unique variability across all features, justifying its separation; (2) classes 1 and 2 share overlapping distributions in key features like discomfort and IMU-recorded motion intensity, with aligned medians and comparable variability, supporting their merger into a single group (class 1); (3) classes 3 and 4 exhibit even greater overlap, with indistinguishable ranges and medians across features such as valence and discomfort. Minor outlier differences do not justify separate treatment, leading to their merger into class 2 (Figure 7).

This merging strategy was applied to state empathy classification, and the results demonstrated the ability to predict state empathy using our method, with the highest balanced accuracy score of 67 ± 1.90% SE achieved using the RF model (Figure 8). This was followed by XGBoost and Logistic Regression with approximately 65% balanced accuracies of ±1.93% SE and ±1.92% SE, respectively (Figure 9). The lowest, but still acceptable, result was obtained using the Decision Tree model, which showed a balanced accuracy of 58 ± 1.88% SE. Additionally, the confusion matrices indicated well-balanced classifications across all models.

#### 5.3.3. State Empathy as Ordinal Classification

For ordinal classification in predicting state empathy, the same classes as for nominal classification and the same window size were used, and the results were similar. The best result was obtained from the model developed using OGB, with a balanced accuracy of 66 ± 1.91% SE. The worst, though still acceptable, result was obtained using the OLR models, with a balanced accuracy of 62 ± 1.95% SE. The results from the Ordinal RF algorithm were 1% worse than the ones obtained with nominal classification (Figure 8 and Figure 9).Compared to earlier studies relying solely on self-reported empathy measures [9,39,47], our integration of physiological signals with ML shows improved predictive performance and provides more objective insights into momentary empathic responses.

### 5.4. Predicting Trait Empathy

#### 5.4.1. Trait Empathy as Regression

The distribution of values for the target variable, trait empathy (affective points), ranged from 5.00 to 8.75. Prior to developing the ML models, all attributes directly related to trait empathy from the QCAE (i.e., perspective-taking, online simulation, emotion contagion, and proximal responsivity) were excluded to prevent data leakage. Additionally, a subsequent experiment was conducted where all questionnaire-obtained data were excluded.

When predicting trait empathy as regression, we achieved moderate success. Since the developed models used not only sensor data but also questionnaire responses as features, and the target value was based on a questionnaire, the inclusion of questionnaire data introduced significant bias, as these responses were self-reported by the participants. The best performance was achieved with the RF model, yielding an MAPE of 9.1 ± 0.32% SEM. This was closely followed by the SVR model with an MAPE of 9.3 ± 0.27% SEM. In contrast, the Decision Tree algorithm performed notably worse, with an MAPE of 12.8 ± 0.33% SEM (Figure 10).

Subsequently, models were developed using only sensor data, excluding questionnaire responses. As anticipated, this omission resulted in higher prediction errors. However, the increase was not as substantial as expected, even though the features used to predict trait empathy differed very little among the first 50 features, suggesting minimal variation in their predictive value. The Elastic Net model produced the best results in this scenario, achieving an MAPE of approximately 10.2 ± 0.29% SEM, followed by the Gradient-Boosting model with an MAPE of 10.4 ± 0.32% SEM. Notably, most models demonstrated errors within a narrow range of 10.4% to 11.5%, except for the Decision Tree model, which, again, exhibited the highest error, with an MAPE of 14.9 ± 0.31% SEM (Figure 11).

These findings partially replicate prior research indicating the moderate predictability of trait empathy from physiological and questionnaire data [55,58], but our results further highlight the potential of sensor-only features to approximate trait empathy with minimal performance loss.

#### 5.4.2. Trait Empathy as Classification

When the ground truth was the trait empathy class, to optimize the classification process, box plots were employed to reduce the number of classes and identify the most effective method for merging them. The plots were generated for the five most important features and the target variable, the trait empathy class. The most important features were identified using RF feature importance. An important note is that the importance of the features when predicting trait empathy differed very little among the first 50 features. However, trait empathy was most strongly influenced by the following features: participants’ ages; the average and minimum values of participants’ state empathy feedback reported after each part of the VR session; and responses regarding immersion in the VR experience and difficulties encountered during the session, based on the Bongiovi Report presence questionnaire. A review of all five box plot graphs suggests that the most consistent merging approach involved merging classes 0 and 1, merging classes 2 and 3, and keeping class 4 separate. This strategy strikes a balance between simplifying the data and preserving key distinctions, guided by the analysis of medians, ranges, and variability across the classes. The reasoning behind this approach is as follows: (1) classes 0 and 1 exhibited similar medians, overlapping ranges, and comparable variability, justifying their merger. Their key features, such as age and average state empathy feedback, showed only minor differences, making them indistinguishable as separate groups; (2) classes 2 and 3 also displayed highly similar distributions, with only slight variations, such as minor differences in the state empathy minimum, and their substantial overlap supports combining them into a single category; (3) class 4 remained distinct, characterized by a broad range, lack of a clear median, and unique variability. This merging strategy simplified classification while preserving essential data patterns (Figure 12).

This merging approach was applied for trait empathy classification, but the results were suboptimal. Balanced accuracy scores for the predictive models ranged from 37% with Logistic Regression to 46% with RF. Furthermore, all developed models exhibited highly imbalanced confusion matrices, with the minority class consistently yielding zero true positives. Consequently, trait empathy prediction in this case could not be effectively addressed as a classification problem.

### 5.5. Statistical Tests

We conducted statistical analyses to examine the influence of various factors on participants’ empathic responses.

#### 5.5.1. Narrative vs. Non-Narrative Versions

Given that our study included five video versions (four narrative and one non-narrative), we employed a one-way Analysis of Variance (ANOVA) to determine whether participants exhibited greater empathy when they were aware of the backstory behind a person’s emotions (narrative videos) compared to when they were only observing the emotions without any contextual narrative (non-narrative video).

The ANOVA results showed that there was no significant difference in state empathy across video groups. The f-statistic, which is defined as the ratio of variance between groups to the variance within groups, was calculated as 0.51, with a corresponding *p*-value of 0.72. The *p*-value represents the probability of obtaining results at least as extreme as the observed results, assuming the null hypothesis is true. Since the *p*-value was greater than 0.05, we concluded that there was no significant difference.

#### 5.5.2. Gender of the Actor

To investigate whether participants were more empathetic toward female or male actors, we conducted an independent-samples *t*-test. This test compared the empathy levels elicited by the two video versions featuring female actors with the two video versions featuring male actors. By classifying the videos into “Female” and “Male” categories, we aimed to assess whether there was a significant difference in empathy based on the gender of the actor.

The *t*-test results showed that there was no significant difference in state empathy between the female and male video versions. This conclusion is based on the T-statistic, which measures the difference between the means of two groups relative to the variation in the data, indicating how much the groups differ in standard error units. In this case, the t-statistic was 1.18, and the *p*-value was 0.24.

#### 5.5.3. Type of Emotion

To investigate whether empathy levels differ across various emotional states represented by video scenes (sad, anxiety, happiness, and anger), we performed an ANOVA. This analysis examined the variance in empathy levels attributable to the video scene (emotional state) while also considering individual differences across participants. The ANOVA results indicated a significant difference in empathy levels between some video versions, with an f-statistic of 25.42 and a very small *p*-value (1.07 × $10^{-14}$), suggesting that the video scene explained a substantial amount of the variance in empathy scores.

Following the ANOVA, we conducted pairwise comparisons using Tukey’s HSD (Honest Significant Difference) post hoc test to identify which specific emotional states evoked significantly different empathy scores. This test compares the differences between each pair of groups while controlling for the family-wise error rate, ensuring that the likelihood of a Type I error is kept at 5% across all comparisons. The family-wise error rate refers to the probability of making one or more Type I errors (false positives) when conducting multiple statistical tests simultaneously.

From the Tukey HSD results, we can see the following:There was a significant difference between “Anger” and “Happiness” (mean difference = 0.8627, *p*-value < 0.05), indicating that participants were more empathetic towards the “Happiness” emotion.Similarly, there was a significant difference between “Anger” and “Sad” (mean difference = 0.3725, *p*-value < 0.05), showing greater empathy towards “Sad” emotions.There was also a significant difference between “Anxiety” and “Happiness” (mean difference = 0.6863, *p*-value < 0.05), where participants were more empathetic towards “Happiness”.There was a significant difference between “Happiness” and “Sad” (mean difference = −0.4902, *p*-value < 0.05), indicating higher empathy towards “Happiness”.However, there was no significant difference between “Anger” and “Anxiety” (mean difference = 0.1765, *p*-value = 0.4595), or between “Anxiety” and “Sad” (mean difference = 0.1961, *p*-value = 0.3639), indicating that empathy levels for these emotions were more similar.

Overall, participants exhibited the highest empathy levels for “Happiness” and “Sad” and the lowest for “Anger”, as indicated by the mean differences and *p*-values from the Tukey HSD test (Table 1). These findings are consistent with prior studies [40,45], confirming that empathy varies with emotional content and that positive and sad emotions elicit stronger empathic responses than anger, highlighting the influence of emotional valence on empathic engagement.

## 6. Discussion and Concluding Remarks

This study introduces an objective and efficient method for predicting both trait and state empathy using physiological responses recorded while participants viewed specially created 3D and 360-degree VR videos. These are the first publicly available VR videos designed explicitly to evoke empathy, created with attention to minimizing biases and subjective influence. Our sensor-based models effectively identified empathic versus non-empathic states, measured general arousal, and distinguished between empathic and non-empathic arousal responses.

Understanding and measuring empathy in this way has wide-ranging implications. In mental health, it offers a framework for identifying empathy deficits and tailoring therapeutic interventions. In VR and empathy training programs—where immersive content is used to challenge perspectives and build empathic skills—our models can guide and assess training effectiveness. In entertainment, the ability to measure empathy in real time enables adaptive VR content that fosters stronger emotional engagement and self-awareness.

Statistical analysis showed that empathy levels were consistent regardless of whether participants understood the causes of others’ emotions and regardless of gender. However, significant differences were found across emotional stimuli: participants expressed higher empathy toward sadness and happiness than toward anger or anxiety. Notably, both narrative and content-free videos elicited comparable empathic responses, further supporting the robustness of our approach.

In summary, this work offers a novel, scalable, and objective framework for empathy prediction, enabling new directions for research and applications across psychology, healthcare, education, and immersive media.

### 6.1. Strengths and Limitations

Through this research, we gained a comprehensive understanding of the outcomes and future directions for advancing empathy research. However, alongside our findings, it is crucial to acknowledge specific limitations.

The sensors used in this study are not practical for everyday use, limiting the immediate applicability of the empathy measurement methods we developed. Nevertheless, they are well-suited for use in VR environments, where measuring empathy is particularly valuable. VR provides an immersive platform for empathy training and measurement. However, a notable limitation of this study is its design, which is more appropriate for laboratory settings than real-world applications, potentially affecting the naturalistic validity of the empathic responses measured.

Despite our efforts to deeply immerse participants and minimize potential disruptions, certain demographic considerations arose. Facial sensors integrated into the VR headset excluded participants who wore glasses, which reduced the representation of older individuals in the sample. As a result, our study primarily included participants aged 19 to 45 years, and the sample was also predominantly female. These demographic imbalances limit the generalizability of our findings, as both age and gender may influence physiological responses and empathic processing. Nonetheless, the relatively large sample size for a VR-based study represents a strength, as it allowed us to train machine learning models with sufficient data to uncover meaningful patterns and provides a strong foundation for future research with more diverse populations.

In statistical modeling, established methodologies can determine the required sample size for model stability. However, for ML models, this calculation is less precise. While we believe the 105 participants included in our study provided a sufficient sample size to develop stable predictive models, we cannot determine with certainty how additional participants might have impacted the results.

The use of convenience sampling, while logistically efficient, presents potential biases, as the sample may not fully represent the broader population. This self-selection bias could limit the generalizability of our findings. However, this approach enabled efficient participant recruitment and extensive data collection in a controlled laboratory environment. Despite these limitations, we believe our study yielded valuable insights into the relationships between empathy, physiological responses, and immersive VR experiences within the scope of our sample.

One significant challenge in this study was establishing a reliable ground truth for measuring empathy. To address this, we used self-reported questionnaires. Although self-assessments come with limitations—such as social desirability bias (participants responding in ways they believe are expected), limited self-awareness, misinterpretation of questions, and fatigue effects from longer forms—they also offer several important advantages. Self-reports provide access to participants’ internal states, offering insight into thoughts and feelings that are not externally observable. Among the available empathy measurement methods, they were the most practical and efficient for our study design: quick to administer and analyze, cost-effective, scalable, and standardized. We selected well-established and validated questionnaires from the literature to ensure reliability and comparability across participants, as all used the same format. Furthermore, since we ultimately grouped participants based on sensor-derived features using box plots, the consistency of the empathy data labels remained unchanged. To reduce the risk of fatigue, we carefully chose short and clear questionnaires. Since participation was anonymous and part of an experiment, there were no expected “correct” answers, reducing pressure on participants. Additionally, we addressed the potential inaccuracy of momentary self-reflection by using two distinct tools: a 3-item state empathy questionnaire and a 31-item trait empathy questionnaire. This dual approach helped reduce bias and encouraged more thoughtful and comprehensive responses.

Statistical analyses were conducted to evaluate empathy across different dimensions. For example, we investigated whether participants exhibited greater empathy for certain emotions and obtained detailed results for all emotions studied. However, these findings may have been influenced by the specific scenes, actors, or performances used in the videos. To reduce this potential bias, we created five different video versions where actors rotated through emotional scenes to ensure a more objective evaluation of empathic responses.

Our findings demonstrated the effective prediction of empathic arousal and the development of predictive models for state and trait empathy based on physiological attributes captured by sensors. However, it is important to note that, while physiological changes were predicted, these changes cannot be definitively attributed to empathy alone, as physiological responses may correlate with, but do not uniquely identify, empathic experiences. The psychological concept of pure empathy, as distinct from empathy mixed with other emotions, remains undefined. To address this complexity, we included non-narrative video versions to reduce the influence of subjective feelings and incorporated an RC (controlled) video segment as a baseline. These measures aimed to minimize errors in empathy prediction and enhance the validity of our results.

### 6.2. Future Work

After building predictive models using standard ML regressors and classifiers, we plan to expand our research by conducting deep learning experiments, such as using end-to-end neural networks, to predict state and trait empathy. Additionally, we plan to extract more features directly from the raw data, moving beyond the affective insights provided by the emteqPRO system. We also intend to make the dataset publicly available, similar to the 3D and 360° videos that are already provided as Appendix A, as it offers a valuable resource for empathy research. Currently, there is a notable lack of comparable datasets and video materials in this field. Our plans fully comply with ethical guidelines, with informed consent obtained from all actors featured in the video recordings and from each participant, ensuring the data will be appropriately anonymized before sharing.

## Figures and Tables

**Figure 1 sensors-25-05766-f001:**
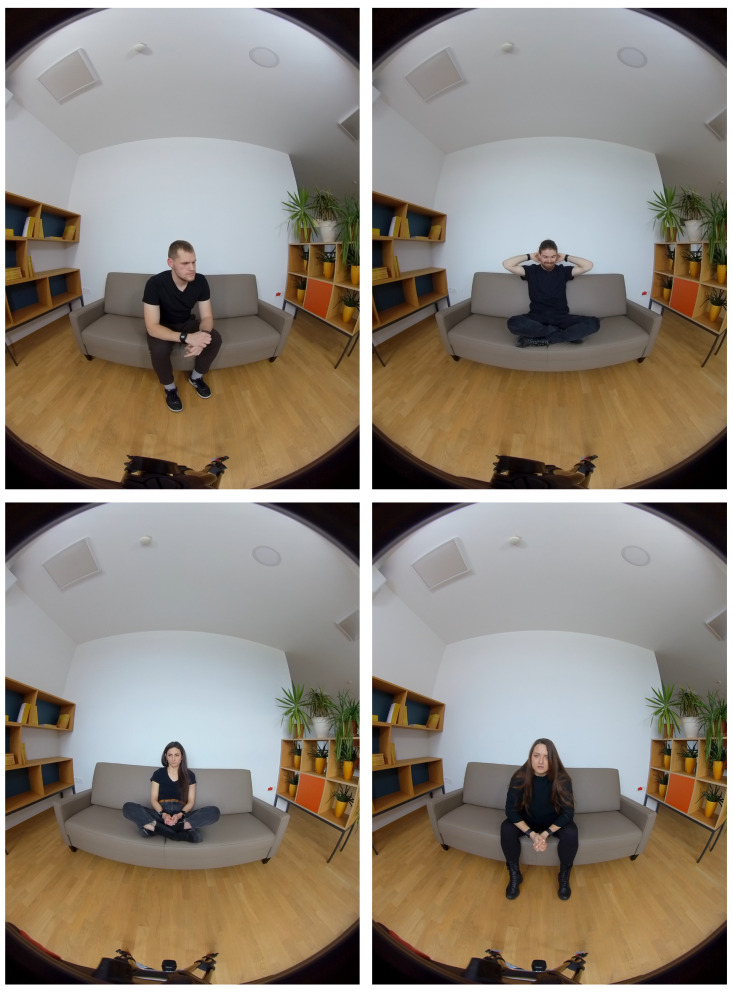
Top to bottom: (**left**) actor one (Dave in the narrative) expressing anger and (**right**) actor two (Jake in the narrative) expressing happiness; (**left**) actress one (Anna) expressing sadness and (**right**) actress two (Leah) expressing anxiousness [53].

**Figure 2 sensors-25-05766-f002:**
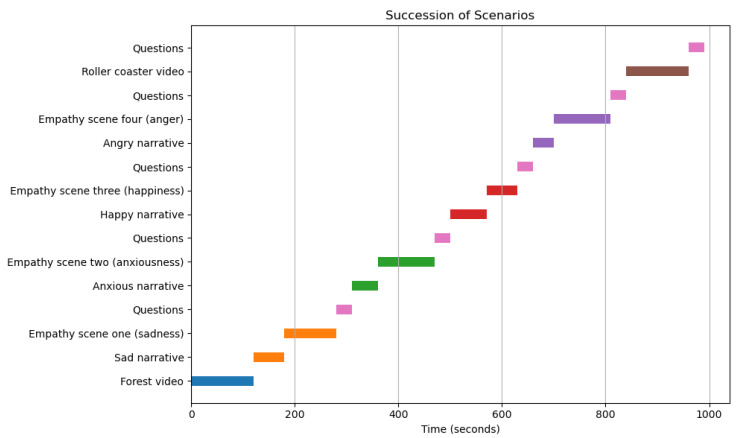
Graphical representation of the sequences of segments in the VR videos (with narratives included).

**Figure 3 sensors-25-05766-f003:**
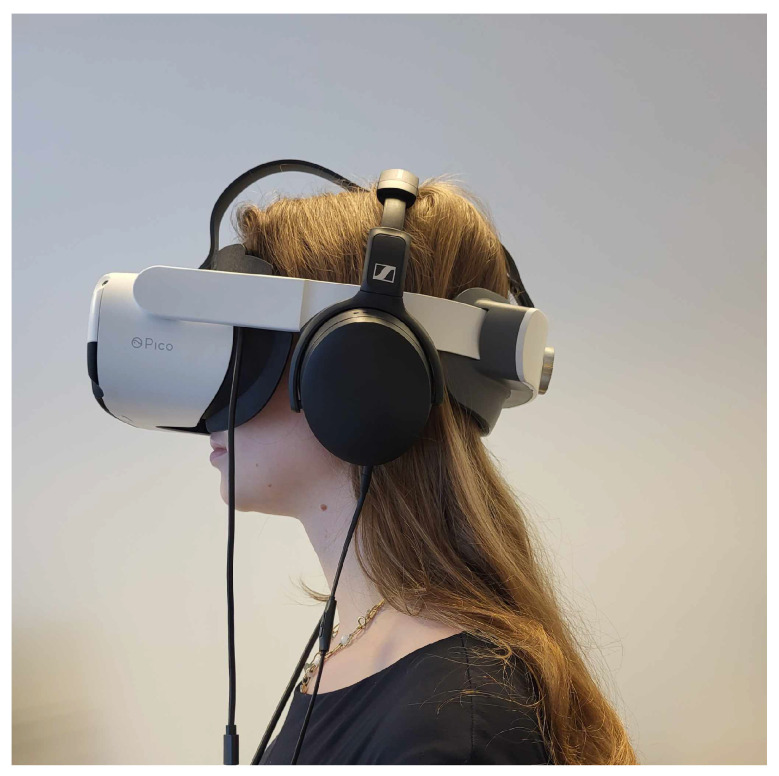
A participant watching videos via the Pico Neo 3 Pro Eye headset in the process of collecting the dataset [53].

**Figure 4 sensors-25-05766-f004:**
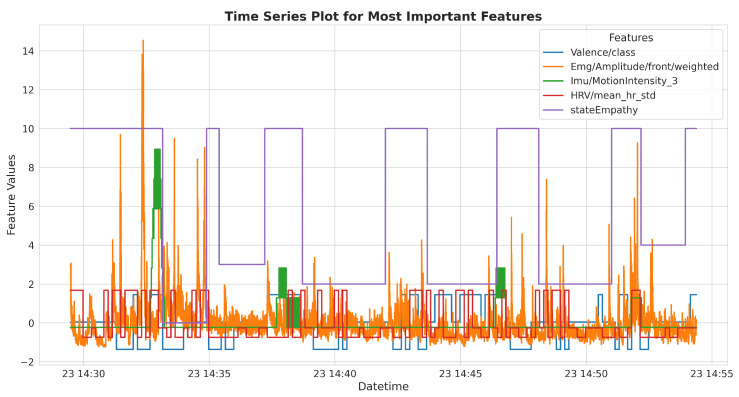
The time series plot for a single participant showing the most important features, including all sequences of the scenarios. The variable stateEmpathy represents the discrete participant responses after each part of the video. Its value is set to 10 before the beginning of the video and during question periods.

**Figure 5 sensors-25-05766-f005:**
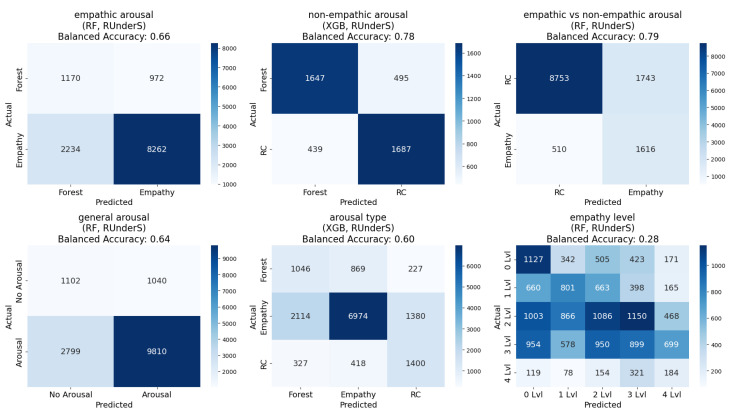
The best confusion matrices for each group of models, developed using a dataset extracted with a 5 s window size and various data-balancing techniques, are shown for the ‘different segments’ and ‘empathy state level’ experiments [53].

**Figure 6 sensors-25-05766-f006:**
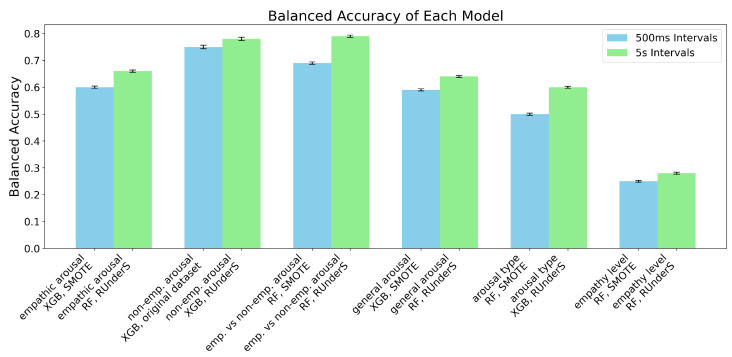
The best accuracies for each group of models, developed using datasets extracted at 500 ms and 5 s intervals with various data-balancing techniques for the ‘different segments’ and ‘empathy state level’ experiment, with error bars representing the standard errors of a proportion of balanced accuracies across participants calculated using an LOSO procedure.

**Figure 7 sensors-25-05766-f007:**
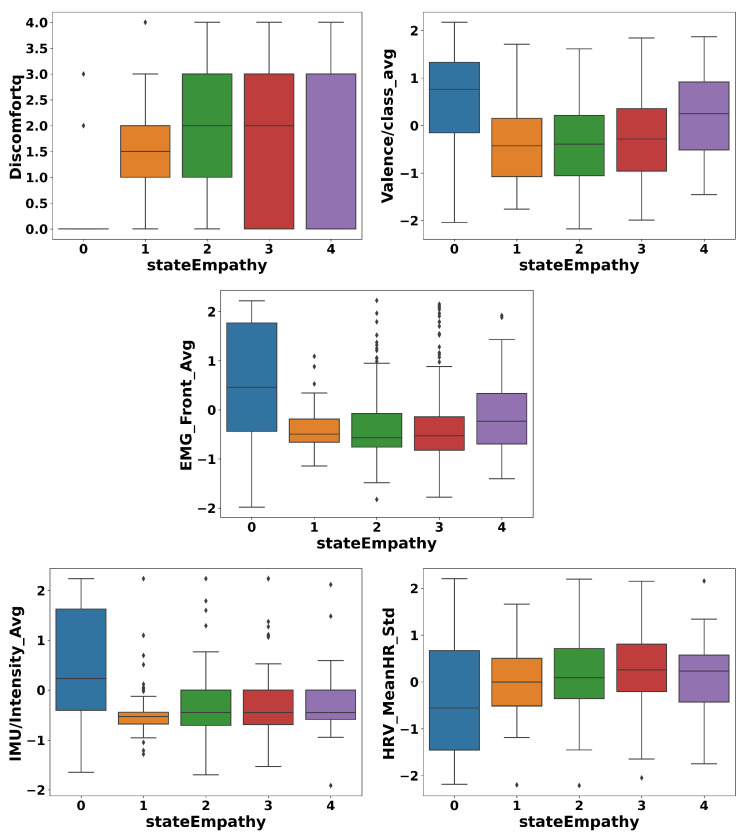
Review of five box plots for the five most important features and the label state empathy, utilized to analyze feature distributions and guide merging decisions.

**Figure 8 sensors-25-05766-f008:**
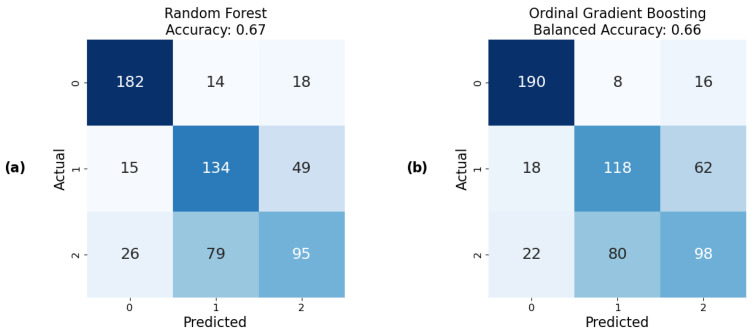
The best confusion matrices and their corresponding accuracies for predicting state empathy obtained using (**a**) nominal classifiers; (**b**) ordinal classifiers.

**Figure 9 sensors-25-05766-f009:**
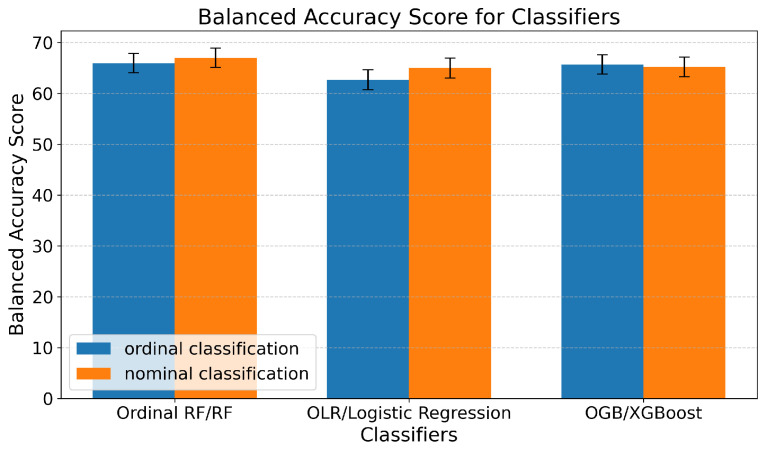
Graphical representation of balanced accuracy for the best predictive models developed using various nominal and ordinal classifiers to predict state empathy, with error bars indicating the standard errors of a proportion across folds.

**Figure 10 sensors-25-05766-f010:**
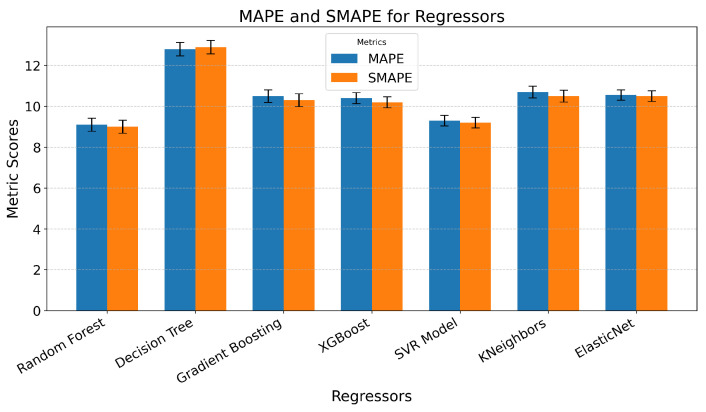
Graphical representation of the MAPE and SMAPE for different predictive models developed using various regressors on sensor and questionnaire data to predict trait empathy, with error bars showing the standard error of the mean across folds.

**Figure 11 sensors-25-05766-f011:**
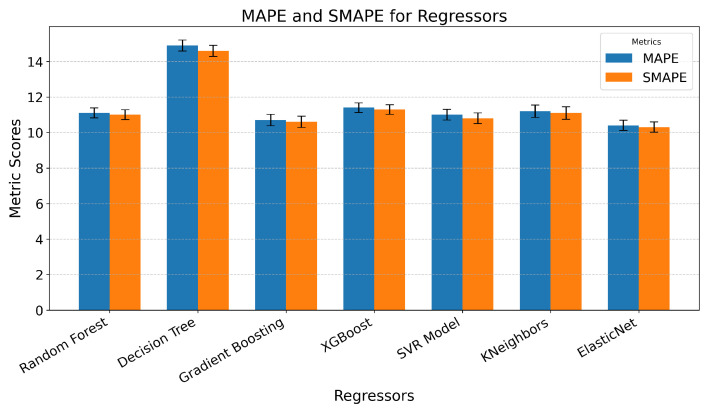
Graphical representation of the MAPE and SMAPE for different predictive models developed using various regressors on sensor data to predict trait empathy, with error bars representing the standard error of the mean across folds.

**Figure 12 sensors-25-05766-f012:**
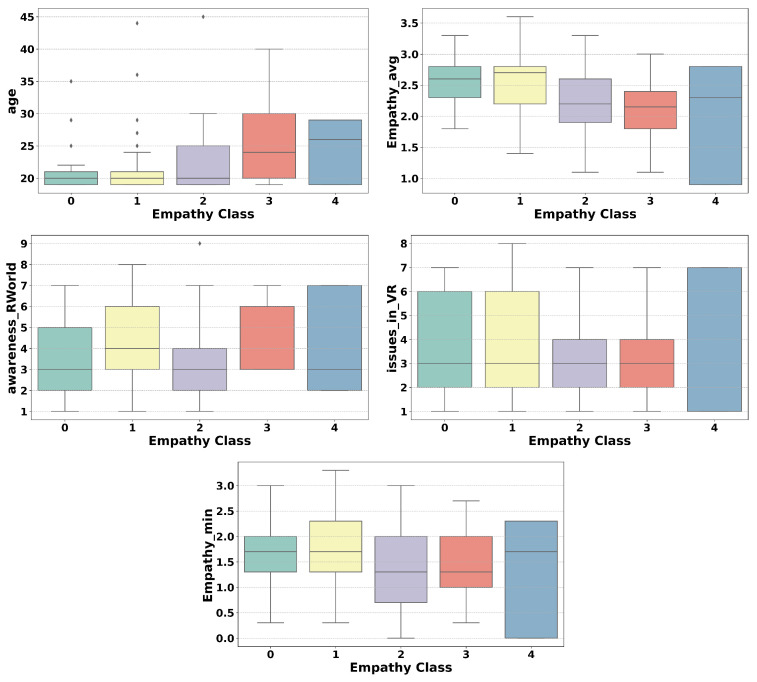
Review of five box plots for the five most important features and the label trait empathy, utilized to analyze feature distributions and guide merging decisions.

**Table 1 sensors-25-05766-t001:** Average state empathy scores among participants for each emotion.

Video Type	Average Empathy Score
Happiness	2.90
Sad	2.41
Anxiety	2.22
Anger	2.04

## Data Availability

The datasets presented in this article are not readily available because the data are part of an ongoing study. Requests to access the datasets should be directed to the corresponding author.

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
