# Peer review of "Predicting Empathy and Other Mental States During VR Sessions Using Sensor Data and Machine Learning"

_sensors, 2025, doi:10.3390/s25185766_

Round 1

Reviewer 1 Report

Comments and Suggestions for Authors

This paper presents an interesting experiment to evaluate whether machine learning models can estimate, from questionnaires and physiological responses, the level of empathy shown by participants during visual stimuli in an immersive virtual environment. The paper is well-structured, albeit somewhat lengthy. It outlines the problem of empathy and how VR is suitable for conducting empathy experiments. An extensive state-of-the-art review is provided before describing the experiments conducted on a large cohort in this work.

The reviewer believes the paper has merit and recommends a series of small adjustments that the authors should make to improve the paper:
- The percentages reported in the abstract should be accompanied by standard deviations.
- When the authors discuss the use of psychophysiological measures, they are referring only to techniques designed to analyze brain behavior. Data-driven approaches based on fuzzy logic have been proposed in the literature to infer psychophysiological states from data collected by wearable systems, which the authors can cite: https://doi.org/10.1186/s12984-024-01501-y, 

What does this point mean? I don't believe that a subject who demonstrates "less empathy" at a given moment requires personalized intervention. The authors should better explain what they mean in point 1 (lines 159-161).
- The legend in figure 4 is illegible. The authors should improve the quality of the figure and the fonts.
- Why did the authors choose to use the models listed in lines 539-541? The authors should justify this choice.
- In addition to the mean value, the authors can also calculate the standard deviation of performance using LOSO (balanced accuracies). It would be appropriate to also represent the deviation of the distribution in Fig. 5.
- In Fig. 7, the authors should also increase the font size.
- In Fig. 8, the authors should rescale the axes between 0 and 1 and add the standard deviation.

Author Response

Comments 1:

- The percentages reported in the abstract should be accompanied by standard deviations.

Response 1: 

We thank the reviewer for this suggestion. We have now updated the abstract so that all percentages are accompanied by their corresponding measures of variability. Specifically, for classification results, we report the Standard Error of the Proportion (SE), and for regression results, we report the Standard Error of the Mean (SEM). This provides a more accurate and consistent representation of variability across folds.

Comments 2:

- When the authors discuss the use of psychophysiological measures, they are referring only to techniques designed to analyze brain behavior. Data-driven approaches based on fuzzy logic have been proposed in the literature to infer psychophysiological states from data collected by wearable systems, which the authors can cite: https://doi.org/10.1186/s12984-024-01501-y, 

Response 2: 

We thank the reviewer for this insightful suggestion. The proposed paper focuses on monitoring physiological signals across multiple sessions to capture the evolution of patients’ emotional and cognitive states, highlighting a shift from initial curiosity and excitement to a more relaxed state. While we have listed methods for empathy measurement, and techniques designed to analyze brain behavior focus specifically on measuring empathy, we will include the proposed paper in the ‘Background and State of the Art’ section to acknowledge its contribution to data-driven psychophysiological state estimation using wearable systems.

Comments 3:

What does this point mean? I don't believe that a subject who demonstrates "less empathy" at a given moment requires personalized intervention. The authors should better explain what they mean in point 1 (lines 159-161).

Response 3: 

We clarified this point by expanding the explanation: the model can assess individuals’ empathy levels across different contexts, providing insight for supporting the development of empathic skills and preventing potential challenges in settings where empathy is critical, such as healthcare or interpersonal relationships. In healthcare in particular, empathy training is an established practice (https://doi.org/10.5116/ijme.61d4.4216).

Comments 4:

- The legend in figure 4 is illegible. The authors should improve the quality of the figure and the fonts.

Response 4: 

We improved the font sizes and overall quality of Figure 4 to ensure the legend and labels are clearly legible.

Comments 5:

- Why did the authors choose to use the models listed in lines 539-541? The authors should justify this choice.

Response 5: 

We thank the reviewer for this comment. We have justified our choice of models in the section indicated by the reviewer, explaining that they were selected to represent a diverse range of methodological approaches (probabilistic, linear, instance-based, tree-based, and ensemble methods), enabling a systematic comparison of their performance.

Comments 6:

- In addition to the mean value, the authors can also calculate the standard deviation of performance using LOSO (balanced accuracies). It would be appropriate to also represent the deviation of the distribution in Fig. 5.

Response 6: 

We thank the reviewer for this suggestion. We have now calculated the SE of the balanced accuracies using the LOSO procedure and included them as error bars in Fig. 5. In addition, we clarified in the Evaluation Metrics section that SE is used for classification results and the SEM is used for regression results, and we provided the corresponding formulas to make our methodology transparent.

Comments 7:

- In Fig. 7, the authors should also increase the font size.

Response 7: 

We increased the font sizes in Figure 7 and in Figure 12, to improve readability as suggested. 

Comments 8:

- In Fig. 8, the authors should rescale the axes between 0 and 1 and add the standard deviation.

Response 8: 

We thank the reviewer for this helpful comment. We have now calculated and reported the Standard Errors for all results, and they are included in all relevant figures, including Fig. 8. The axes in Fig. 8 have also been rescaled, and the error bars now reflect the corresponding Standard Errors.

The manuscript with track changes can be found in the attachment.

Reviewer 2 Report

Comments and Suggestions for Authors

The study aims to objectively predict empathy levels in VR using machine learning models trained on physiological and self-reported data.

The paper presents a strong methodological approach to measuring empathy through VR and physiological sensors, with several strengths that make it suitable for publication, though some revisions would strengthen it further:

The abstract could better clarify the sample size and demographics of participants and highlight the study’s contribution.

The introduction is well-referenced. I noticed an excessive background detail on empathy definitions and VR applications. However, less emphasis is given to the research gap. I would also suggest highlighting early that this study uniquely combines VR with physiological sensing and machine learning to predict empathy. I would also suggest better associating the physiological measures with empathy’s cognitive/affective components to justify their selection over other modalities.

The following section provides a comprehensive overview of VR-based empathy research, measurement methods, and ML applications. However, it is overly detailed and occasionally repetitive, which risks obscuring the main research gap. The literature review would benefit from tighter synthesis, with clearer transitions linking each subsection to the study’s specific contributions. Greater focus should be placed on highlighting the novelty of integrating VR, sensor-based physiological data, and ML for empathy prediction, as this is the unique contribution of the present work.

In the methodology, I would suggest better justifying the sample size using, for instance, power analysis or comparable studies. I would also suggest addressing gender imbalance and discussing its potential impact on generalizability. Exclusion criteria must be better explained.

The methodology is robust. I would suggest better addressing potential multicollinearity among physiological features (e.g., HRV and EMG correlations) that may inflate model performance. It would be helpful to discuss the ecological validity of lab-controlled VR stimuli versus real-world empathy elicitation.

The discussion effectively summarizes key findings. Please clarify causal limitations (i.e., physiological responses may correlate with but not uniquely identify empathy). In addition, expand ethical considerations for public dataset release.

Author Response

Comments 1:

The abstract could better clarify the sample size and demographics of participants and highlight the study’s contribution.

Response 1: 

We thank the reviewer for this suggestion. We have revised the abstract to include the sample size and demographic details of the participants, and we have also clarified the study’s contribution by explicitly highlighting how our work advances empathy research and provides valuable resources for future studies.

Comments 2:

The introduction is well-referenced. I noticed an excessive background detail on empathy definitions and VR applications. However, less emphasis is given to the research gap. I would also suggest highlighting early that this study uniquely combines VR with physiological sensing and machine learning to predict empathy. I would also suggest better associating the physiological measures with empathy’s cognitive/affective components to justify their selection over other modalities.

Response 2: 

We thank the reviewer for the constructive feedback. In response, we made the following revisions to strengthen the introduction: (1) we added a clear statement of the research gap by emphasizing the limited integration of physiological sensing with immersive VR for modeling empathic processes; (2) we highlighted earlier in the introduction that this study uniquely combines immersive VR, physiological sensing, and machine learning to predict empathy; and (3) we provided a stronger justification for our choice of physiological measures by explicitly linking PPG and EMG to affective arousal, and IMU to cognitive engagement, thereby aligning them with empathy’s multidimensional components. While we have described in detail the rationale for selecting these sensors and their association with empathy in our protocol study (https://doi.org/10.1371/journal.pone.0307385), we also included a concise explanation here to clarify their relevance in the current work.

Comments 3:

The following section provides a comprehensive overview of VR-based empathy research, measurement methods, and ML applications. However, it is overly detailed and occasionally repetitive, which risks obscuring the main research gap. The literature review would benefit from tighter synthesis, with clearer transitions linking each subsection to the study’s specific contributions. Greater focus should be placed on highlighting the novelty of integrating VR, sensor-based physiological data, and ML for empathy prediction, as this is the unique contribution of the present work.

Response 3: 

We thank the reviewer for this insightful comment. We have revised the background section to group repetitive descriptions by themes (e.g., healthcare, prosocial behavior, technical factors), added synthesis sentences to clarify transitions, and emphasized early the novelty of integrating immersive VR with multimodal physiological sensing and machine learning to predict empathy. This maintains the detailed examples while highlighting the study’s unique contribution.

Comments 4:

In the methodology, I would suggest better justifying the sample size using, for instance, power analysis or comparable studies. I would also suggest addressing gender imbalance and discussing its potential impact on generalizability. Exclusion criteria must be better explained.

Response 4: 

We thank the reviewer for these suggestions. The justification for the sample size was previously detailed in our protocol paper and is now included in the manuscript, referencing comparable studies from our narrative review (Kizhevska, E., Ferreira-Brito, F., Guerreiro, T. J., & Lustrek, M. (2022). Using Virtual Reality to Elicit Empathy: a Narrative Review. VR4Health@MUM, 19–22.) to support its adequacy for developing stable predictive models. Gender imbalance is acknowledged in the ‘Materials and Data Collection Process’ section, and we have added a discussion of its potential impact on generalizability. Additionally, the exclusion criteria have been clarified, specifying the health conditions and factors considered for participant eligibility.

Comments 5:

The methodology is robust. I would suggest better addressing potential multicollinearity among physiological features (e.g., HRV and EMG correlations) that may inflate model performance. It would be helpful to discuss the ecological validity of lab-controlled VR stimuli versus real-world empathy elicitation.

Response 5: 

We thank the reviewer for this suggestion. In the first wave of our experiments, reported in a short conference paper (Kizhevska, E., & Luštrek, M. (2024). Predicting Mental States During VR Sessions Using Sensor Data and Machine Learning.), we evaluated potential multicollinearity among physiological features using a correlation matrix and did not observe any substantial multicollinearity. We did not include these details in the current mavnuscript to avoid repetition of previously published material, but we are happy to include them if deemed necessary. Additionally, our use of leave-one-subject-out (LOSO) cross-validation helps prevent inflation of model performance due to correlated features. If anything, collinearity could contribute to overfitting and thus decrease performance. We understand that collinearity can sometimes inflate performance when using the R2 performance measure, but we do not use it. Regarding ecological validity, we acknowledge the limitations of lab-controlled VR stimuli compared to real-world empathy elicitation, and this point is discussed in the “Strengths and Limitations” subsection of the Discussion.

Comments 6:

The discussion effectively summarizes key findings. Please clarify causal limitations (i.e., physiological responses may correlate with but not uniquely identify empathy). In addition, expand ethical considerations for public dataset release.

Response 6: 

As suggested by the reviewer, we have added a clarification of the causal limitations in the Strengths and Limitations subsection of the Discussion, highlighting that physiological responses may correlate with but do not uniquely identify empathy. Furthermore, we have expanded the ethical considerations regarding the release of the dataset in the ‘Participants and Recording Setup’ subsection of the ‘Materials and Data Collection Process’.

The manuscript with track changes can be found in the attachment.

Reviewer 3 Report

Comments and Suggestions for Authors
  1. It is unscientific to use a lot of descriptive language in the introduction section instead of clearly pointing out existing research gaps
  2. The author did not demonstrate the basic framework of the relevant algorithm, nor did they derive the relevant formulas, which is inappropriate
  3. The author did not compare their research results with existing literature and lacked critical thinking

Author Response

Comments 1:

It is unscientific to use a lot of descriptive language in the introduction section instead of clearly pointing out existing research gaps

Response 1: 

We thank the reviewer for this valuable comment and agree with the observation. In the revised version of the manuscript, we added a clearer explanation of the existing research gap. Specifically, while there is growing interest in both VR applications and empathy research, there remains a notable gap in integrating physiological sensing with immersive VR to objectively model empathic processes. Our study directly addresses this gap by exploring the relationship between different types of empathy and physiological responses during immersive VR experiences.

Comments 2:

The author did not demonstrate the basic framework of the relevant algorithm, nor did they derive the relevant formulas, which is inappropriate

Response 2: 

We acknowledge the reviewer’s comment regarding the derivation of algorithmic formulas. The machine learning models and evaluation metrics used in this study are standard methods, and detailed mathematical formulations are provided in the cited references. Given the focus of this paper on VR-based empathy prediction, we opted to reference these well-established methods rather than reproduce the formulas, in order to maintain clarity and conciseness. Our primary contribution lies in presenting new materials (360º 3D VR videos), a novel dataset, the methodology we applied, experimental results, and insights, as well as the application of machine learning models to this unique dataset for the purpose of predicting empathic responses in immersive VR scenarios. This emphasis on the dataset, experimental design, and empirical findings is the core novelty of our work, rather than the development of new algorithms.

But while we opted not to reproduce the well-known formulas to maintain clarity and conciseness, we would be happy to include such descriptions in future work and, of course, welcome any specific suggestions from the reviewer on how best to present them.

Comments 3:

The author did not compare their research results with existing literature and lacked critical thinking

Response 3: 

Direct comparisons with existing literature are possible when one is tackling a well researched problem, especially if benchmark datasets are available. In our paper neither is the case. We did our best to include some comparisons with existing literature in the Results subsections for each experiment. These comparisons highlight how our findings relate to prior VR, physiological, and machine learning studies on empathy. While our research primarily focuses on automating empathy assessment and establishing a standardized approach to complement or improve existing questionnaire-based methods—thereby reducing subjectivity and potential biases, as explained in the Introduction and Discussion sections—we have also clarified certain points in the Discussion to better express our critical thinking and interpretation of the results.

The manuscript with track changes can be found in the attachment.

Round 2

Reviewer 3 Report

Comments and Suggestions for Authors

The author has made significant revisions, although the article lacks innovation, it does not affect its publication